# Analysis of IMRT patient specific quality assurance using EPID measurement data from the Halcyon linear accelerator

Shinhaeng Cho☯, Jea-Uk Jeong☯, Yong Hyub Kim, Mee Sun Yoon, Taek-Keun Nam, Sung-Ja Ahn, Ju-Young Song ⓘ *

Department Radiation Oncology, Chonnam National University Medical School, Gwangju, Republic of Korea

☯ These authors contributed equally to this work.
* jysong@jnu.ac.kr

## Abstract

This study investigated the characteristics of electronic portal imaging device (EPID)-based perpendicular field-by-field patient-specific quality assurance (PFF-PSQA) on the Halcyon linear accelerator by comparing them with the true composite (TC)-PSQA performed using the Delta$^4$ Phantom+ (ScandiDos, Sweden) system. Both methods demonstrated excellent dose agreement under the AAPM TG-218 gamma criteria and strong correlations were observed between their results across prostate, nasopharynx, and breast cases. However, gamma passing rate (GPR) reductions due to treatment isocenter displacements were more pronounced in TC-PSQA, whereas EPID-based PFF-PSQA consistently showed relatively higher and more stable GPR values. A two-way repeated measures ANOVA analysis confirmed a significant difference in sensitivity to displacement-induced errors, supporting the clinical robustness of EPID-based PFF-PSQA. In addition, EPID-based PFF-PSQA yielded tolerance and action limits of 99.5% and 99.4%, respectively, which exceed the conventional 95% and 90% thresholds specified in the report of American association of physicists in medicine (AAPM) TG-218, highlighting the necessity of refining evaluation criteria for this approach. Overall, these findings indicate that EPID-based PFF-PSQA can serve as a reliable and clinically alternative to TC-PSQA.

## Introduction

Intensity modulated radiation therapy (IMRT), which enables the delivery of a highly conformal prescription dose distribution to the tumor target while minimizing dose to surrounding normal tissues, requires verification of dosimetric accuracy by measuring the actual dose distribution delivered by the treatment machine and comparing it with the calculated distribution in the treatment planning system (TPS) before starting patient treatment. This verification procedure is referred to as IMRT patient-specific

**Data availability statement:** All relevant data are within the manuscript and its Supporting Information files.

**Funding:** The author(s) received no specific funding for this work.

**Competing interests:** The authors have declared that no competing interests exist.

quality assurance (PSQA). In the early stages of IMRT implementation, PSQA was primarily performed by measuring doses with film and ionization chambers [1,2]. However, due to a time-consuming and single-use consumable in these methods, dose distribution measurements are now predominantly conducted using diode and ionization chamber array devices [3–6]. Various research has been conducted on IMRT PSQA methods, and studies have been established to propose appropriate evaluation criteria for PSQA [7, 8].

One of PSQA methods is to measure the dose distribution using an electric portal imaging device (EPID) installed in a linear accelerator and many studies have been conducted in this regard [9]. Although this approach allows only perpendicular field-by-field (PFF) measurements and cannot analyze the true composite (TC) dose distribution of all treatment beams used in IMRT, it is widely employed in clinical practice because it does not require the setup of additional QA devices and allows for relatively straightforward comparison and analysis with treatment planning results. In particular, the Halcyon (Varian Medical Systems, USA) linear accelerator (LINAC), with its O-ring design, has the advantage that treatment beams are consistently measured by a fixed EPID. As a result, the utilization of EPID measurement data for dose verification is more convenient compared to conventional C-arm linear accelerators, in which the EPID operates on a robotic arm.

In this study, we aimed to evaluate whether there is a significant difference between IMRT PSQA results obtained using the EPID of the Halcyon linear accelerator and the true composite dose analysis results obtained with conventional detector array devices. Through this approach, we attempted to validate the feasibility of EPID-based PSQA for IMRT using Halcyon LINAC and to establish reference standards for clinical application by calculating the tolerance limits (TL) and action limits (AL) which serve as the evaluation criteria for PSQA results.

## Materials and methods

### Preparation of IMRT PSQA plans with the Halcyon LINAC

Test IMRT plans were generated to analyze the characteristics of PSQA results using the EPID of the Halcyon LINAC. In this study, IMRT plans were generated for a total of 30 patients, including 10 with prostate cancer, 10 with nasopharynx cancer, and 10 with breast cancer, using the volumetric modulated arc therapy (VMAT) technique. This study conducted PSQA for VMAT plans instead of fixed-beam IMRT, reflecting the widespread clinical use of VMAT-based IMRT. The intention was to evaluate PFF-PSQA results under dynamic arc conditions, where the gantry rotates during irradiation, rather than under static fixed-angle measurements. VMAT plans were generated using two full arcs for prostate cancer, four full arcs for brain cancer, and four partial arcs for breast cancer patients. The VMAT plans were created using the Eclipse 16.1 (Varian Medical Systems, USA) planning system.

For each VMAT plan, two types of PSQA plans were generated. One was based on the conventional TC dose verification method with a diode detector array, employing the Delta$^4$ Phantom+ (ScandiDos, Sweden) system, while the other was generated using the PFF dose measurement method with an EPID and subsequently

analyzed through the portal dosimetry 16.1 software (Varian Medical Systems, USA). The portal dose was calculated using the anisotropic analytical algorithm (AAA).

The Delta⁴ Phantom+ system has a central region with a 5 mm resolution and an outer region with a 10 mm resolution, with a 0.05 mm thickness for each p-type Si diode. The panel of Halcyon EPID is 43 cm × 43 cm with 1280 × 1280 pixel matrix which makes the spatial resolution 2.98 mm$^{-1}$ and a high-resolution pixel size of 0.34 mm.

## PSQA measurements and dose accuracy evaluation

In this study, measurements were performed under three different conditions for each established PSQA plan. First, measurements were carried out at the treatment isocenter as defined in the treatment plan to analyze the accuracy differences between the two PSQA methods.

Next, to evaluate the variations in the accuracy analysis results between the two methods under beam delivery errors, measurements were performed by introducing intentional isocenter shifts. Specifically, separate measurements and analyses were performed with the treatment isocenter displacements of 2 mm in both the lateral and longitudinal directions, as well as for displacements of 3 mm. In the TC measurement method with the Delta⁴ Phantom+ system, positional errors were introduced by adjusting the treatment couch position, whereas in the PFF measurement method with the EPID, the measured dose distributions were translated within the portal dosimetry software to simulate equivalent positional errors. Since all planning parameters were identical to those of the original plan and only the isocenter was shifted, each arc yielded an equivalent dose distribution that was spatially translated according to the applied shift.

All the dosimetric accuracy in this study was evaluated with the gamma passing rate (GPR) calculated in the gamma evaluation, using the absolute dose difference and distance-to-agreement criteria, which were 3%/2 mm and 2%/2 mm, respectively with low-dose cutoff threshold 10% of the maximum dose. For both PSQA methods, the alignment function was not applied, and the global normalization method was used.

We evaluated the correlation of GPR values calculated by the two PSQA methods with respect to treatment isocenter displacements and analyzed the changes in mean GPR values as the magnitude of displacement increased. A two-way repeated measures ANOVA was performed to evaluate the significant differences in GPR between the two PSQA methods, resulting from dose deviations caused by isocenter shifts.

## Tolerance limits (TL) and action limits (AL) calculation

The TL and AL, which serve as the criteria for determining whether the PSQA results pass or fail and for taking action in case of failure, were calculated for each method to serve as institutional guidelines for corrective measures in case of failure. In this study, a total of 30 original PSQA data points—10 measurements for each case—obtained without any isocenter shift were used in the calculations. The calculation of TL and AL was based on the AAPM TG-218 [10], as described by the following equations.

$$TL = \bar{x} - 2.660 \times \frac{1}{n-1} \sum_{i=2}^{n} \left| x_i - x_{i-1} \right|$$

and

$$AL = 100\% - \frac{\beta \times \sqrt{\left(\sigma^2 + \left(\bar{x} - T\right)^2\right)}}{2}$$

Where $\sigma^2$ and $\bar{x}$ are the variance and mean value of the calculated GPR, $x_i$ is the result of each GPR, $n$ is the total number of calculated GPR, $T$ is 100% (the target value of the GPR), and constant $\beta$ is 6.0 [11].

## Results and discussion

Figs 1 and 2 show typical examples of the PSQA results obtained in this study. The dose errors caused by treatment iso-center displacements in TC-PSQA using the Delta⁴ Phantom+ system can be observed in Fig 1(b) and the corresponding dose errors in PFF-PSQA using EPID measurement data can be seen in Fig 2(b).

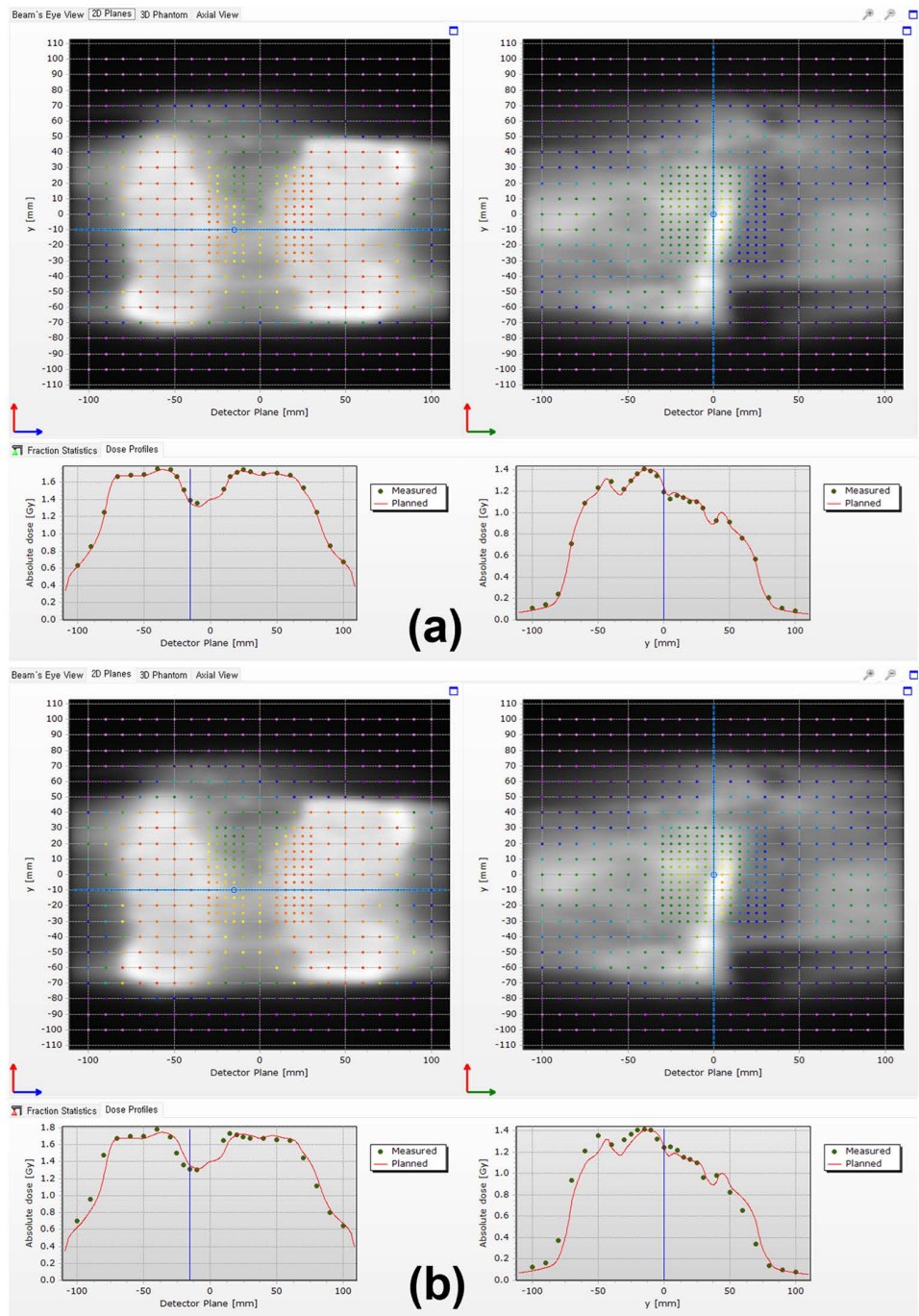

**Fig 1. Example of the TC-Delta⁴ PSQA results in N2 patient.** The difference between the calculated dose in the plan and measured dose in a original isocenter (GPR: 100.0% with 3%/2 mm criteria) and b isocenter displacement with 3 mm:3 mm (GPR: 66.4% with 3%/2 mm criteria).

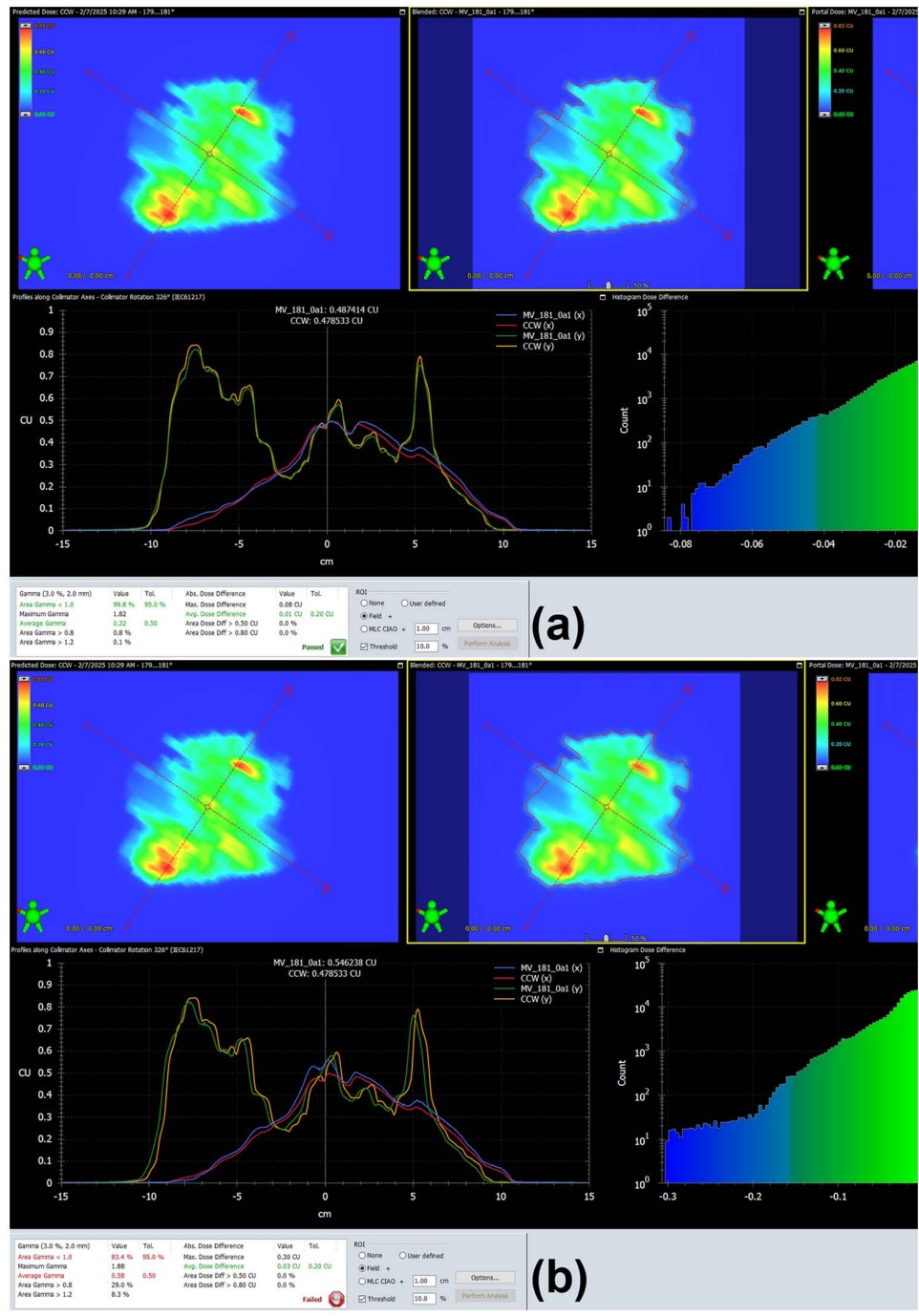

**Fig 2. Example of the PFF-EPID PSQA results in N2 patient.** The difference between the calculated dose in the plan and measured dose in **a** original isocenter (GPR: 99.8% with 3%/2 mm criteria) and **b** isocenter displacement with 3 mm:3 mm (GPR: 83.4% with 3%/2 mm criteria).

Table 1 summarizes the changes in GPR values resulting from treatment isocenter displacements in TC-PSQA with the Delta⁴ Phantom+ system, while Table 2 provides the corresponding results for PFF-PSQA using EPID portal dosimetry. In Tables 1 and 2, the notation "2mm:2mm" indicates that the isocenter was shifted by 2 mm in both the lateral and

**Table 1. GPR (%) variations according to displacements of the treatment isocenter in TC-PSQA using the Delta [4] Phantom+ system.**

| Prostate | 3%/2 mm criteria | | | 2%/2 mm criteria | | |
|---|---|---|---|---|---|---|
| | Original | 2mm:2 mm | 3mm:3 mm | Original | 2mm:2 mm | 3mm:3 mm |
| P1 | 99.9 | 87.1 | 75.2 | 97.7 | 79.7 | 64.6 |
| P2 | 99.1 | 87.8 | 63.9 | 94.7 | 78.0 | 53.6 |
| P3 | 99.3 | 87.4 | 69.1 | 94.1 | 79.1 | 60.1 |
| P4 | 100.0 | 86.7 | 68.0 | 96.9 | 76.3 | 57.3 |
| P5 | 98.8 | 84.9 | 57.6 | 93.8 | 73.8 | 47.1 |
| P6 | 99.4 | 76.2 | 69.0 | 97.1 | 61.0 | 55.4 |
| P7 | 100.0 | 78.3 | 66.7 | 97.1 | 66.4 | 53.4 |
| P8 | 100.0 | 81.6 | 69.6 | 97.2 | 69.3 | 59.7 |
| P9 | 99.9 | 83.0 | 71.3 | 98.0 | 69.8 | 57.2 |
| P10 | 99.9 | 80.2 | 69.9 | 97.2 | 64.4 | 56.4 |
| Average | 99.6 | 83.3 | 68.0 | 96.4 | 71.8 | 56.5 |
| Nasopharynx | 3%/2 mm criteria | | | 2%/2 mm criteria | | |
| | Original | 2mm:2 mm | 3mm:3 mm | Original | 2mm:2 mm | 3mm:3 mm |
| N1 | 99.8 | 96.1 | 85.4 | 99.5 | 94.3 | 79.5 |
| N2 | 100.0 | 81.3 | 66.4 | 99.6 | 70.4 | 53.5 |
| N3 | 99.2 | 80.9 | 70.4 | 96.3 | 71.9 | 59.1 |
| N4 | 99.4 | 87.5 | 79.8 | 98.3 | 77.8 | 68.3 |
| N5 | 99.3 | 84.0 | 72.8 | 96.8 | 75.2 | 61.4 |
| N6 | 99.9 | 83.5 | 74.9 | 99.2 | 72.8 | 64.4 |
| N7 | 99.0 | 85.6 | 77.5 | 95.1 | 77.3 | 67.2 |
| N8 | 99.7 | 79.8 | 62.2 | 99.1 | 68.2 | 53.2 |
| N9 | 98.7 | 95.8 | 84.7 | 97.0 | 89.0 | 75.2 |
| N10 | 99.4 | 84.9 | 74.2 | 96.8 | 75.6 | 63.9 |
| Average | 99.4 | 85.9 | 74.8 | 97.8 | 77.3 | 64.6 |
| Breast | 3%/2 mm criteria | | | 2%/2 mm criteria | | |
| | Original | 2mm:2 mm | 3mm:3 mm | Original | 2mm:2 mm | 3mm:3 mm |
| B1 | 99.8 | 91.0 | 79.8 | 97.7 | 84.6 | 70.4 |
| B2 | 100.0 | 79.5 | 60.5 | 98.3 | 70.2 | 47.8 |
| B3 | 99.4 | 84.3 | 59.4 | 96.3 | 76.4 | 50.3 |
| B4 | 99.6 | 85.9 | 69.1 | 98.2 | 78.8 | 59.6 |
| B5 | 99.8 | 91.9 | 82.2 | 98.1 | 87.1 | 75.8 |
| B6 | 100.0 | 85.8 | 74.3 | 97.2 | 80.1 | 66.7 |
| B7 | 99.6 | 88.0 | 76.0 | 97.0 | 81.0 | 65.0 |
| B8 | 100.0 | 87.4 | 72.1 | 97.6 | 79.8 | 62.2 |
| B9 | 100.0 | 85.3 | 60.7 | 98.6 | 77.8 | 49.9 |
| B10 | 100.0 | 78.8 | 71.0 | 99.0 | 69.7 | 64.2 |
| Average | 99.8 | 85.8 | 70.5 | 97.8 | 78.6 | 61.2 |

longitudinal directions, while "3mm:3mm" likewise represents a 3 mm shift in both directions. The GPR values in Table 2 represent the mean GPRs of the individual fields that constitute each patient's VMAT plan. Based on the AAPM TG-218 recommended 3%/2 mm criteria applied in our institution for gamma evaluation, the original PSQA without isocenter variation demonstrated excellent dose agreement in both TC-PSQA with Delta[4] and PFF-PSQA with EPID, yielding average GPR values of 99.6% and 99.9%, respectively, with no notable difference between the two methods. The

**Table 2. GPR (%) variations according to displacements of the treatment isocenter in PFF-PSQA using the EPID portal dosimetry.**

| Prostate | 3%/2 mm criteria | | | 2%/2 mm criteria | | |
|---|---|---|---|---|---|---|
| | Original | 2 mm:2 mm | 3 mm:3 mm | Original | 2 mm:2 mm | 3 mm:3 mm |
| P1 | 99.9 | 99.2 | 89.7 | 99.3 | 97.3 | 84.1 |
| P2 | 100.0 | 97.9 | 85.3 | 99.1 | 94.9 | 78.6 |
| P3 | 99.9 | 98.7 | 88.5 | 98.6 | 95.3 | 80.5 |
| P4 | 100.0 | 98.9 | 89.9 | 99.2 | 97.1 | 84.0 |
| P5 | 99.9 | 98.2 | 86.8 | 98.8 | 95.1 | 79.7 |
| P6 | 99.9 | 98.6 | 87.3 | 98.7 | 95.5 | 79.5 |
| P7 | 100.0 | 98.6 | 86.2 | 99.2 | 96.1 | 80.2 |
| P8 | 99.7 | 97.1 | 86.7 | 98.3 | 94.4 | 81.1 |
| P9 | 99.8 | 97.8 | 89.7 | 98.5 | 94.7 | 84.6 |
| P10 | 99.9 | 99.1 | 90.3 | 99.0 | 96.8 | 83.5 |
| Average | 99.9 | 98.4 | 88.0 | 98.9 | 95.7 | 81.6 |
| **Nasopharynx** | **3%/2 mm criteria** | | | **2%/2 mm criteria** | | |
| | Original | 2 mm:2 mm | 3 mm:3 mm | Original | 2 mm:2 mm | 3 mm:3 mm |
| N1 | 100.0 | 99.4 | 89.3 | 99.9 | 98.7 | 84.4 |
| N2 | 99.9 | 98.6 | 89.1 | 99.7 | 97.1 | 83.1 |
| N3 | 100.0 | 98.9 | 89.1 | 99.7 | 97.6 | 83.4 |
| N4 | 100.0 | 99.2 | 90.6 | 99.8 | 97.9 | 85.6 |
| N5 | 100.0 | 98.8 | 89.9 | 99.7 | 97.4 | 84.9 |
| N6 | 100.0 | 99.0 | 88.9 | 99.8 | 97.8 | 83.8 |
| N7 | 99.9 | 98.8 | 88.7 | 99.5 | 97.3 | 83.3 |
| N8 | 99.9 | 98.7 | 89.4 | 99.5 | 97.2 | 84.0 |
| N9 | 100.0 | 99.1 | 89.6 | 99.8 | 98.0 | 84.5 |
| N10 | 100.0 | 98.9 | 89.9 | 99.9 | 97.5 | 84.2 |
| Average | 100.0 | 98.9 | 89.4 | 99.7 | 97.6 | 84.1 |
| **Breast** | **3%/2 mm criteria** | | | **2%/2 mm criteria** | | |
| | Original | 2 mm:2 mm | 3 mm:3 mm | Original | 2 mm:2 mm | 3 mm:3 mm |
| B1 | 99.9 | 99.2 | 89.9 | 97.6 | 95.1 | 83.4 |
| B2 | 99.7 | 95.1 | 81.8 | 97.0 | 89.4 | 74.3 |
| B3 | 99.7 | 98.4 | 84.3 | 96.2 | 93.4 | 77.6 |
| B4 | 99.8 | 98.3 | 85.3 | 96.5 | 93.8 | 79.0 |
| B5 | 99.6 | 94.9 | 82.9 | 97.8 | 90.4 | 75.9 |
| B6 | 99.8 | 98.1 | 86.6 | 97.2 | 94.1 | 79.4 |
| B7 | 99.6 | 94.3 | 80.0 | 95.9 | 89.2 | 74.4 |
| B8 | 99.9 | 99.0 | 86.2 | 97.6 | 95.2 | 78.6 |
| B9 | 99.4 | 93.2 | 76.8 | 97.1 | 93.1 | 81.3 |
| B10 | 100.0 | 99.7 | 90.3 | 99.2 | 98.1 | 85.4 |
| Average | 99.9 | 99.2 | 89.9 | 97.2 | 93.2 | 78.9 |

results indicated that larger isocenter displacements led to greater dose errors and reduced GPR values in both PSQA methods.

Fig 3 illustrates the correlation of GPR values calculated by the two PSQA methods according to treatment isocenter displacements. The calculated Pearson correlation coefficients were 0.89 and 0.86 for prostate cases, 0.80 and 0.79 for

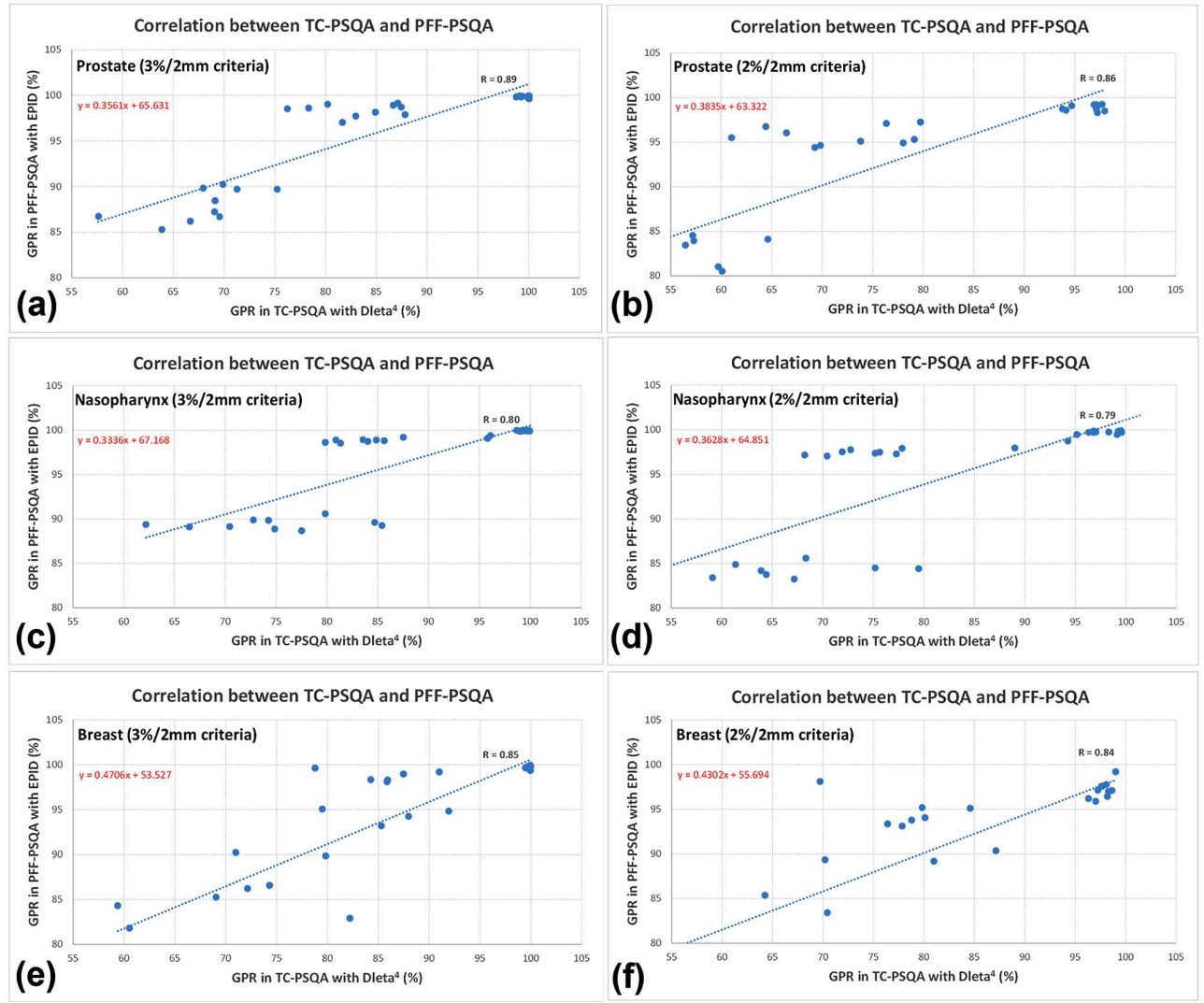

**Fig 3. Correlation of GPR between TC-Delta⁴ PSQA and PFF-EPID PSQA in accordance with isocenter displacement.** a prostate (3%/2 mm criteria), b prostate (2%/2 mm criteria), c nasopharynx (3%/2 mm criteria), d nasopharynx (2%/2 mm criteria), e breast (3%/2 mm criteria) and f breast (2%/2 mm criteria).

nasopharynx cases, and 0.85 and 0.84 for breast cases under the gamma evaluation criteria of 3%/2 mm and 2%/2 mm, respectively, confirming a strong correlation between the results obtained from the two PSQA methods. Although the extent of reduction differed, both PSQA methods demonstrated a decrease in GPR values as IMRT dose errors increased, indicating that substituting one method for the other would not lead to different trends in PSQA analysis results.

Fig 4 shows the trend of decreasing average GPR values for each case according to the magnitude of isocenter displacement. In both PSQA methods, GPR values were found to decrease as the displacement increased, and this trend was more pronounced when gamma evaluation was performed with the 2%/2 mm criteria. Overall, the GPR values obtained with PFF-PSQA using EPID were higher than those with TC-PSQA using Delta⁴ Phantom+ system, and even in the presence of dose errors induced by isocenter displacements, the PFF-PSQA using EPID method consistently demonstrated relatively higher GPR values. This tendency of EPID-based PFF-PSQA to be less affected

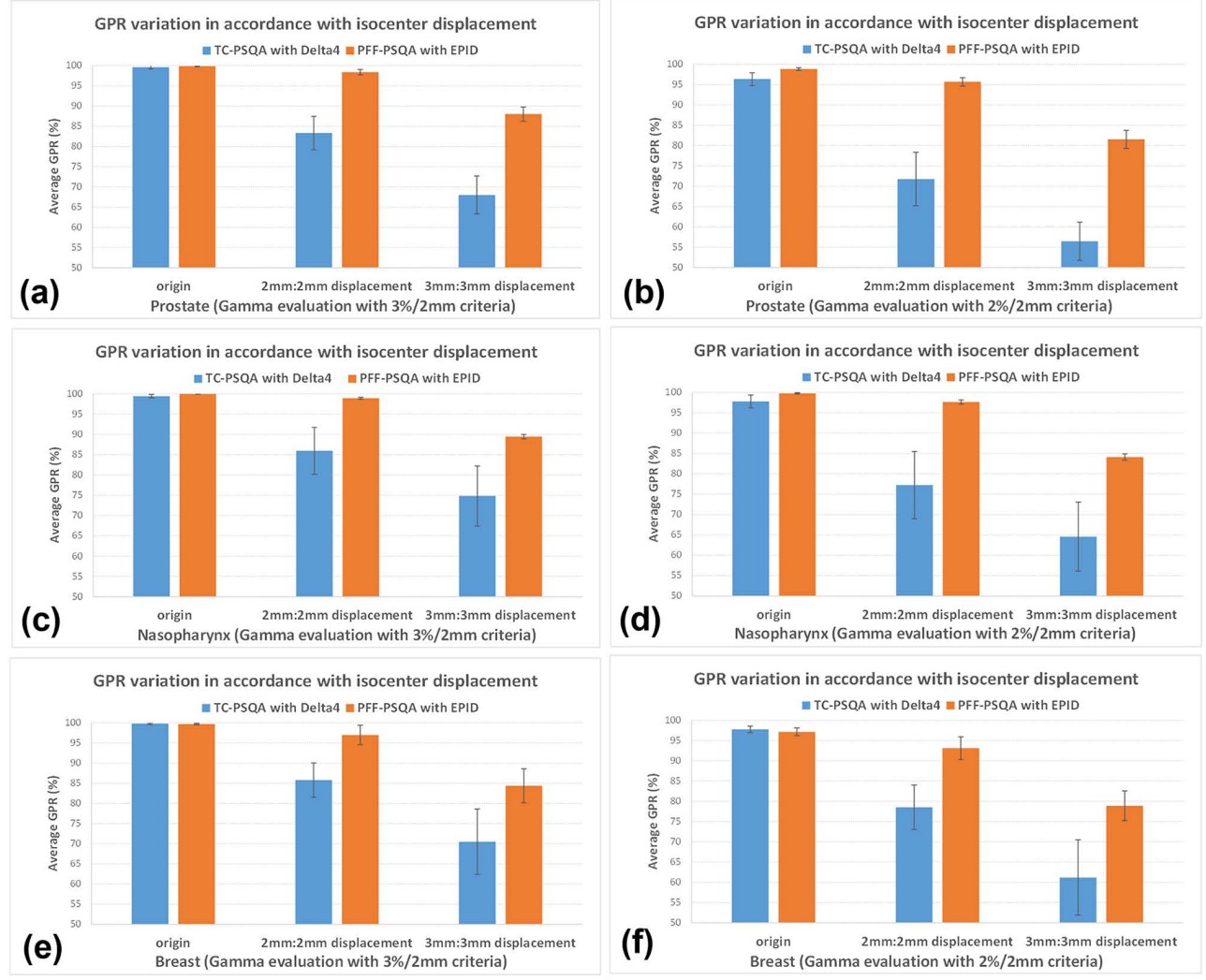

**Fig 4. Variation of average GPR values in accordance with isocenter displacement.** a prostate (3%/2 mm criteria), b prostate (2%/2 mm criteria), c nasopharynx (3%/2 mm criteria), d nasopharynx (2%/2 mm criteria), e breast (3%/2 mm criteria) and f breast (2%/2 mm criteria).

by dose error factors and to provide more stable, higher GPR values has also been consistently observed in other studies [12, 13]. The relatively higher GPR values observed with EPID-based PFF-PSQA may be attributed to the absence of setup errors that can occur when positioning a separate detector array device on the treatment couch in TC-PSQA. In addition, because the beam is always incident perpendicularly on the EPID, the influence of beam angle on the detector can be eliminated, thereby providing more stable and consistently higher GPR values. From a general perspective, the use of EPID, which has relatively superior detector resolution, would be expected to yield lower GPR values, as it can more sensitively reflect subtle dose discrepancies. However, in this study, when comparing the results of TC-PSQA and PFF-PSQA performed using different measurement approaches, the results showed an opposite tendency. This may be attributed to the fact that the variation in dose distribution for each arc caused by isocenter displacement was less sensitive than the overall dose distribution change measured in TC-PSQA. Therefore, to accurately and robustly assess the influence of detector resolution, it is essential to conduct comparisons by employing a

uniform PSQA approach (such as PFF-PSQA or TC-PSQA) and evaluating the outcomes based on the degree of dose gradient variation.

Fig 5 presents the average GPR reduction according to increasing treatment isocenter displacements, demonstrating that the TC-PSQA method using Delta⁴ exhibited a greater decrease compared to the PPF-PSQA method using EPID. Both methods showed a decrease in GPR as the displacement error increased; however, the patterns of reduction differed between the two methods. The difference in sensitivity was particularly evident in the significant PSQA method × displacement error interaction, indicating that one method was more sensitive to variations in dose error than the other. The two-way repeated measures ANOVA analysis revealed a highly significant difference between the two PSQA methods, with an *F*-value of 134.9 and a *p*-value below 0.0001, indicating that the degree of GPR reduction with increasing isocenter displacement differed substantially between them. Therefore, in clinical practice where a PSQA method that is less sensitive to dose error factors and provides reliable stability is required, EPID-based PFF-PSQA may represent the more appropriate choice. Recently, most radiotherapy treatments have been delivered using IMRT techniques, leading to an excessive increase in the workload associated with IMRT PSQA. Although institutions with a limited number of treatment units and sufficient QA personnel are able to perform PSQA without major difficulties, facilities operating multiple treatment units with insufficient staffing often experience significant challenges in maintaining efficient PSQA processes. Under such circumstances, the EPID-based PFF-PSQA method may provide a more stable and efficient alternative, owing to its simplified workflow and reduced dependence on physical measurement devices. This approach is particularly advantageous in situations where treatment plans are generated shortly before patient treatment and limited time is available for QA measurement, analysis, and corrective actions, thereby supporting more consistent and reliable QA performance under time-constrained clinical conditions.

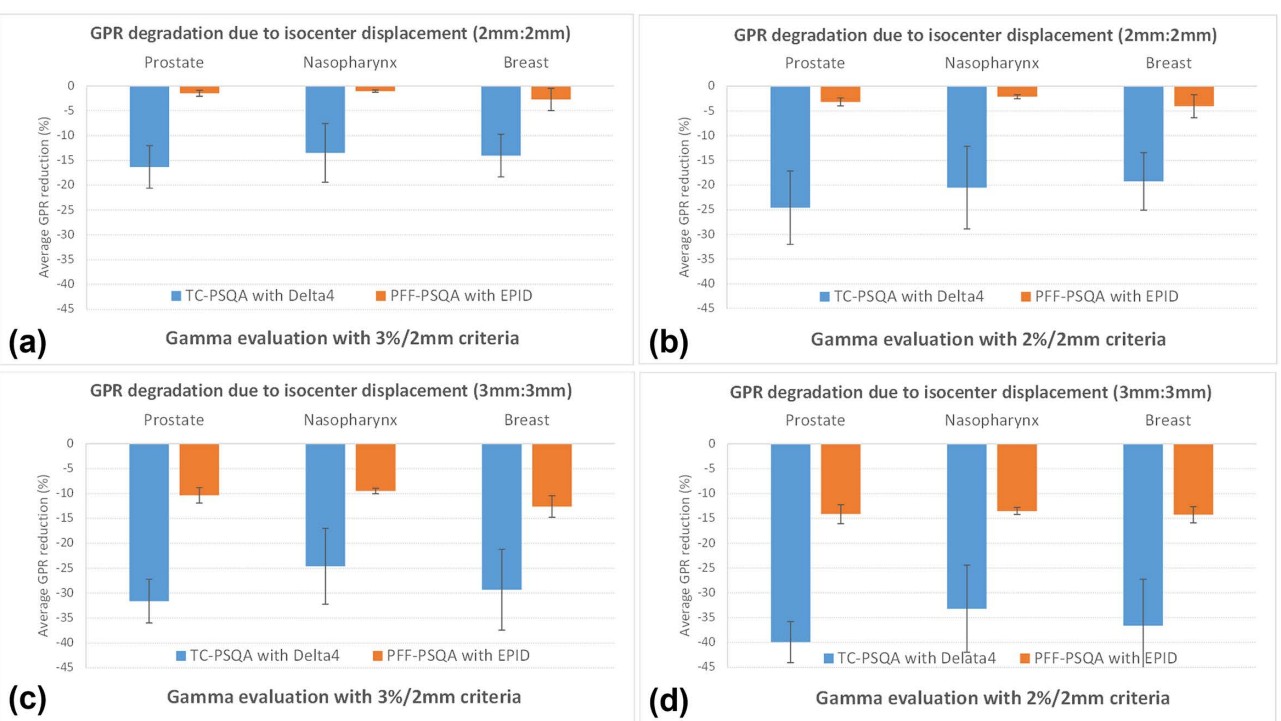

**Fig 5. Comparison of GPR reduction due to treatment isocenter displacements.** Displacements of 2 mm:2 mm (**a** 3%/2 mm criteria, **b** 2%/2 mm criteria) and 3 mm:3 mm (**c** 3%/2 mm criteria, **d** 2%/2 mm criteria).

Based on the results of the two PSQA methods performed in this study, the tolerance and action limits for evaluating the IMRT QA outcomes of each method were calculated according to the AAPM TG-218 guidelines, as presented in Table 3. Both the TL and AL values derived from the GPR calculated with the 3%/2 mm criteria recommended in AAPM TG-218 exceeded 98% in the two PSQA methods, which were higher than the recommended thresholds of 95% and 90%, respectively. Notably, the EPID-based PFF-PSQA demonstrated TL and AL values of 99.5% and 99.4%, indicating that the current criteria may underestimate its clinical performance. As summarized in Table 3, adopting the conventional 95% and 90% thresholds for EPID-based PFF-PSQA would be more clinically appropriate if the gamma evaluation criteria were set to 2%/2 mm instead of 3%/2 mm. This finding underscores the need to optimize tolerance and action limits to better reflect the clinical applicability of EPID-based PFF-PSQA methods. The TL and AL values obtained in this study were derived from a limited dataset of 30 samples performed without isocenter displacement, which may restrict their generalizability. Therefore, these values will be continuously refined and updated using additional measurement data collected in the future.

In the PSQA results of IMRT using the EPID of the Halcyon LINAC analyzed in this study, a strong positive correlation was observed with the TC-PSQA results obtained using the Delta[4] diode detector array, confirming that EPID-based PFF-PSQA can serve as a viable alternative for IMRT PSQA. However, considering that EPID-based PFF-PSQA tends to yield higher GPR values and is less sensitive to dose error factors than conventional TC-PSQA, further analyses are needed to determine appropriate TL and AL criteria for evaluating its results.

In this study, we analyzed only the use of EPID measurement data for PFF-PSQA; however, recent studies have increasingly investigated methods to calculate dose distributions within the patient CT using EPID data acquired from beams transmitted through the patient [14–16]. Through EPID-based transit dosimetry using the back-projection approach, the characteristics of TC-PSQA can also be realized. Therefore, the use of EPID measurement data is expected to be increasingly applied to IMRT PSQA as a method that integrates the advantages of both PFF-PSQA and TC-PSQA. Although AAPM TG-218 recommends the TC-PSQA method over the PFF-PSQA method as the preferred approach for IMRT PSQA due to its inherent advantages, it is considered that the EPID-based PSQA method could reasonably serve as an alternative to conventional TC-PSQA when the patient dose distribution reconstructed from transit dosimetry is utilized in conjunction with the PFF analysis results. In the future, we plan to further conduct research in this area to establish a systematic and comprehensive IMRT PSQA protocol utilizing EPID on the Halcyon LINAC.

## Conclusions

This study demonstrated that EPID-based PFF-PSQA on the Halcyon LINAC shows strong correlation with TC-PSQA using the Delta[4] Phantom+ system, while consistently yielding higher and more stable GPR values and reduced sensitivity to isocenter displacement errors. Tolerance and action limits derived from EPID-based PFF-PSQA exceeded conventional AAPM TG-218 thresholds, highlighting the need for optimized evaluation criteria. These results suggest that EPID-based PFF-PSQA on the Halcyon LINAC is a reliable and clinically alternative to TC-PSQA, with potential for further development through EPID-based transit dosimetry.

Table 3. Tolerance limits (TL) and action limits (AL) calculated in accordance with AAPM TG-218 (AAPM recommended universal TL: 95%, AL: 90%, with 3%/2 mm and a10% dose threshold).

|  | 3%/2 mm criteria | | 2%/2 mm criteria | |
|  | TC-PSQA (Delta[4]) | PFF-PSQA (EPID) | TC-PSQA (Delta[4]) | PFF-PSQA (EPID) |
|---|---|---|---|---|
| TL | 98.5% | 99.5% | 93.2% | 97.0% |
| AL | 98.4% | 99.4% | 90.8% | 94.4% |

## Supporting information

**S1 File. GPR variation in prostate cases.**
(XLSX)

**S2 File. GPR variation in nasopharynx cases.**
(XLSX)

**S3 File. GPR variation in breast cases.**
(XLSX)

**S4 File. Analysis of correlation between TC-PSQA and PFF-PSQA.**
(XLSX)

**S5 File. Calculation of TL (Tolerance Limit) and AL (Action Limit).**
(XLSX)

**S6 File. Calculation of GPR degradation due to isocenter displacement.**
(XLSX)

## Author contributions

**Conceptualization:** Shinhaeng Cho, Jea-Uk Jeong, Ju-Young Song.

**Data curation:** Shinhaeng Cho, Yong Hyub Kim, Mee Sun Yoon, Taek-Keun Nam, Sung-Ja Ahn.

**Formal analysis:** Jea-Uk Jeong.

**Funding acquisition:** Ju-Young Song.

**Investigation:** Shinhaeng Cho, Yong Hyub Kim, Mee Sun Yoon, Taek-Keun Nam, Sung-Ja Ahn.

**Methodology:** Shinhaeng Cho, Jea-Uk Jeong, Yong Hyub Kim, Mee Sun Yoon.

**Project administration:** Ju-Young Song.

**Resources:** Shinhaeng Cho, Jea-Uk Jeong.

**Validation:** Shinhaeng Cho, Ju-Young Song.

**Visualization:** Shinhaeng Cho.

**Writing – original draft:** Shinhaeng Cho, Jea-Uk Jeong.

**Writing – review & editing:** Ju-Young Song.

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
