## [Decision Letter · Decision Letter 0]

1 Apr 2026

PONE-D-25-66579Analysis of IMRT patient specific quality assurance using EPID measurement data from the Halcyon linear acceleratorPLOS One

Dear Dr. Song,

Thank you for submitting your manuscript to PLOS ONE. After careful consideration, we feel that it has merit but does not fully meet PLOS ONE’s publication criteria as it currently stands. Therefore, we invite you to submit a revised version of the manuscript that addresses the points raised during the review process.

We look forward to receiving your revised manuscript.

Kind regards,

Moshi Geso, Ph.D.

Academic Editor

PLOS One

Journal Requirements:

Additional Editor Comments:

Dear authors

Please note the comments raised by the reviewers and correct the manuscript according before submitting for publication

Reviewers' comments:

Reviewer's Responses to Questions

**Comments to the Author**

1. Is the manuscript technically sound, and do the data support the conclusions?

Reviewer #1: Yes

Reviewer #2: Yes

2. Has the statistical analysis been performed appropriately and rigorously? 

Reviewer #1: Yes

Reviewer #2: Yes

3. Have the authors made all data underlying the findings in their manuscript fully available?

Reviewer #1: Yes

Reviewer #2: Yes

4. Is the manuscript presented in an intelligible fashion and written in standard English?

Reviewer #1: Yes

Reviewer #2: Yes

5. Review Comments to the Author

Reviewer #1: This study is an important investigation about the patient specific quality assurance using EPID data from Halcyon Linac. As the clinical usage of these comparatively new Linacs are increasing, this paper is practically important. The authors have covered all the necessary points aimed for this investigation. Data and the statistical analysis clearly support the conclusion s. Authors made the data fully available without restrictions. The manuscript is well structured and fits the guideline of the journal. If the figures meet the criteria, this manuscript is acceptable for publication.

Reviewer #2: The manuscript is technically sound, well written and easy to understand.

The topic is highly relevant to PSQA and clearly demonstrates the robustness of EPID based dosimetry. (We have seen similar results in our department).

The axis titles in Figure 3 are hard to read.

The bar graph titles in Figures 4 & 5 are too small to read.

6. PLOS authors have the option to publish the peer review history of their article (what does this mean?). If published, this will include your full peer review and any attached files.

Reviewer #1: No

Reviewer #2: **Yes:**Frank Gagliardi

---

## [Author Response · Author response to Decision Letter 1]

15 Apr 2026

Reviewer #1: This study is an important investigation about the patient specific quality assurance using EPID data from Halcyon Linac. As the clinical usage of these comparatively new Linacs are increasing, this paper is practically important. The authors have covered all the necessary points aimed for this investigation. Data and the statistical analysis clearly support the conclusion s. Authors made the data fully available without restrictions. The manuscript is well structured and fits the guideline of the journal. If the figures meet the criteria, this manuscript is acceptable for publication.

Thank you for your positive evaluation and for the favorable decision regarding the publication of our manuscript. We have thoroughly reviewed the journal’s guidelines for figure files and have resubmitted the revised versions to ensure full compliance. We sincerely appreciate your constructive feedback and support.

Reviewer #2: The manuscript is technically sound, well written and easy to understand.

The topic is highly relevant to PSQA and clearly demonstrates the robustness of EPID based dosimetry. (We have seen similar results in our department).

The axis titles in Figure 3 are hard to read.

The bar graph titles in Figures 4 & 5 are too small to read.

Thank you for your valuable review and suggestions. As requested, we have revised the figure files to improve their clarity and readability. The specific modifications are as follows:

- Figure 3: The font size of the axis titles has been increased and bolded to enhance legibility.

- Figures 4 & 5: The titles of the bar graphs have been made larger and bolder to ensure they are easier to read.

We have resubmitted these updated files for your reconsideration.

---

## [Editor Report · Decision Letter 1]

20 Apr 2026

Analysis of IMRT patient specific quality assurance using EPID measurement data from the Halcyon linear accelerator

PONE-D-25-66579R1

Dear Dr. Song,

We’re pleased to inform you that your manuscript has been judged scientifically suitable for publication and will be formally accepted for publication once it meets all outstanding technical requirements.

Kind regards,

Moshi Geso, Ph.D.

Academic Editor

PLOS One
---

## [Editor Report · Acceptance letter]

PONE-D-25-66579R1

PLOS One

Dear Dr. Song,

I'm pleased to inform you that your manuscript has been deemed suitable for publication in PLOS One. Congratulations! Your manuscript is now being handed over to our production team.

Kind regards,

on behalf of

A/Professor Moshi Geso

Academic Editor

PLOS One